META-RESEARCH ARTICLE

# A survey of biomedical journals to detect editorial bias and nepotistic behavior

**Alexandre Scanff** [1], **Florian Naudet** [1], **Ioana A. Cristea** [2], **David Moher** [3,4], **Dorothy V. M. Bishop** [5], **Clara Locher** [1] *

1 Univ Rennes, CHU Rennes, Inserm, CIC 1414 (Centre d'Investigation Clinique de Rennes), Rennes, France, 2 Department of Brain and Behavioral Sciences, University of Pavia, Pavia, Italy, 3 Centre for Journalology, Clinical Epidemiology Program, Ottawa Hospital Research Institute, Ottawa, Ontario, Canada, 4 School of Epidemiology and Public Health, University of Ottawa, Ottawa, Ontario, Canada, 5 Department of Experimental Psychology, University of Oxford, Oxford, United Kingdom

* clara.locher@univ-rennes1.fr

**Data Availability Statement:** All files are available from the Open Science Framework (https://osf.io/6e3uf/).

## Abstract

Alongside the growing concerns regarding predatory journal growth, other questionable editorial practices have gained visibility recently. Among them, we explored the usefulness of the Percentage of Papers by the Most Prolific author (PPMP) and the Gini index (level of inequality in the distribution of authorship among authors) as tools to identify journals that may show favoritism in accepting articles by specific authors. We examined whether the PPMP, complemented by the Gini index, could be useful for identifying cases of potential editorial bias, using all articles in a sample of 5,468 biomedical journals indexed in the National Library of Medicine. For articles published between 2015 and 2019, the median PPMP was 2.9%, and 5% of journal exhibited a PPMP of 10.6% or more. Among the journals with the highest PPMP or Gini index values, where a few authors were responsible for a disproportionate number of publications, a random sample was manually examined, revealing that the most prolific author was part of the editorial board in 60 cases (61%). The papers by the most prolific authors were more likely to be accepted for publication within 3 weeks of their submission. Results of analysis on a subset of articles, excluding nonresearch articles, were consistent with those of the principal analysis. In most journals, publications are distributed across a large number of authors. Our results reveal a subset of journals where a few authors, often members of the editorial board, were responsible for a disproportionate number of publications. To enhance trust in their practices, journals need to be transparent about their editorial and peer review practices.

## Introduction

Research integrity matters across the research ecosystem. In this process, scientific journal editors are key actors that ensure the trustworthiness of the scientific publication process. But, paraphrasing Dr. Drummond Rennie's famous quote, who is guarding those guardians? [1] Some of our team (CL, IC, DM, and FN) had doubts that anyone does such safe guarding in the case of New Microbes and New Infections (NMNI), an Elsevier journal, whose most

**Funding:** The authors received no specific funding for this work.

**Competing interests:** The authors have declared that no competing interests exist.

**Abbreviations:** MeSH, Medical Subject Headings; NCBI, National Center for Biotechnology Information; NLM, National Library of Medicine; NMNI, New Microbes and New Infections; PPMP, Percentage of Papers by the Most Prolific author; WoS, Web of Science.

prolific author, Didier Raoult, coauthored 32% of its 728 published papers [2]. NMNI's editor-in-chief and 6 additional associate editors of the journal work directly for, and report to, Raoult. Together, these editors authored 44% of the 728 papers published in the journal as of June 25, 2020. We suggested that such "self-promotion journals" were "a new type of illegitimate publishing entity, which could have certain key characteristics such as (i) a constantly high proportion of papers published by the same group of authors, (ii) relationships between the editors and these authors, and (iii) publication of low-quality research" [2]. We applied a preliminary approach to detect these "self-promotion journals" in the field of infectious disease using a measure easy to compute: the proportion of contributions published in a journal by the most prolific author, i.e., the one who published the most articles in a given time period [2]. In journals publishing more than 50 papers over 5 years, it was rare to see journals where a specific author published more than 10% of the papers, and, indeed, NMNI was a clear outlier. Note, however, this is a crude measure as it is based on all published articles, whatever their type (research, letter, editorial, etc.) and therefore may give high scores for legitimate contributions by active editors.

Coincidently, one of our colleagues (DB) reported a similar analysis for the addiction subfield of psychology in a blog post [3]. This analysis focused on research articles only. She found a bimodal distribution of "the percentage by the most prolific" measure, identical to that observed for NMNI, with only 3 out of 99 journals having a score over 8%. In 2 of these journals, the high score was attributable to the same individual, who was on the editorial board of the journal, and who had published together with the editor-in-chief. Bishop also noted that the same method identified a journal editor, Johnny Matson, who had been found previously to be publishing copiously in journals edited by himself or other editors [3]. Furthermore, for many of these papers, indirect evidence of superficial or absent peer reviews was suspected because of the remarkably rapid turnaround, often within a week or less, between the dates recorded for submission and manuscript acceptance. This circumstantial evidence of unethical editorial practice can only be obtained, however, in journals that report these dates for published manuscripts.

These convergent analyses from different fields suggest that the Percentage of Papers by the Most Prolific author (PPMP) deserves consideration as a potential red flag to identify journals that are suspected of biased editorial decision-making—what we now term "nepotistic journals."

We may draw parallels between the PPMP and studies on resource distribution in economics. A highly prolific author who is an outlier on the PPMP measure in effect monopolizes a large proportion of a journal's publications. This analogy supports another, more complex measure, the Gini index [4], used in econometrics to describe resource distribution inequalities. This measure was recently applied in bibliometrics to explore inequality in authorship across authors publishing in high-impact academic medical journals [5]. Applied to our context, it could be used to quantify imbalances in the patterns of authorship within a journal: the values of the Gini index range from 0 (perfect equality in numbers of articles among authors) to 1 (major inequality).

We set out to apply these 2 indices to a very large dataset of biomedical journals over a 5-year time frame, to describe outliers using these indices, and to describe time intervals between submission and acceptance dates as a potential indicator of unfair or partisan editorial practices.

## Results

### Journal selection and description

Using the search query on the United States National Library of Medicine (NLM) catalog, 11,665 journals labeled with at least one of 152 "Broad Subject Terms" were retrieved. **Fig 1**

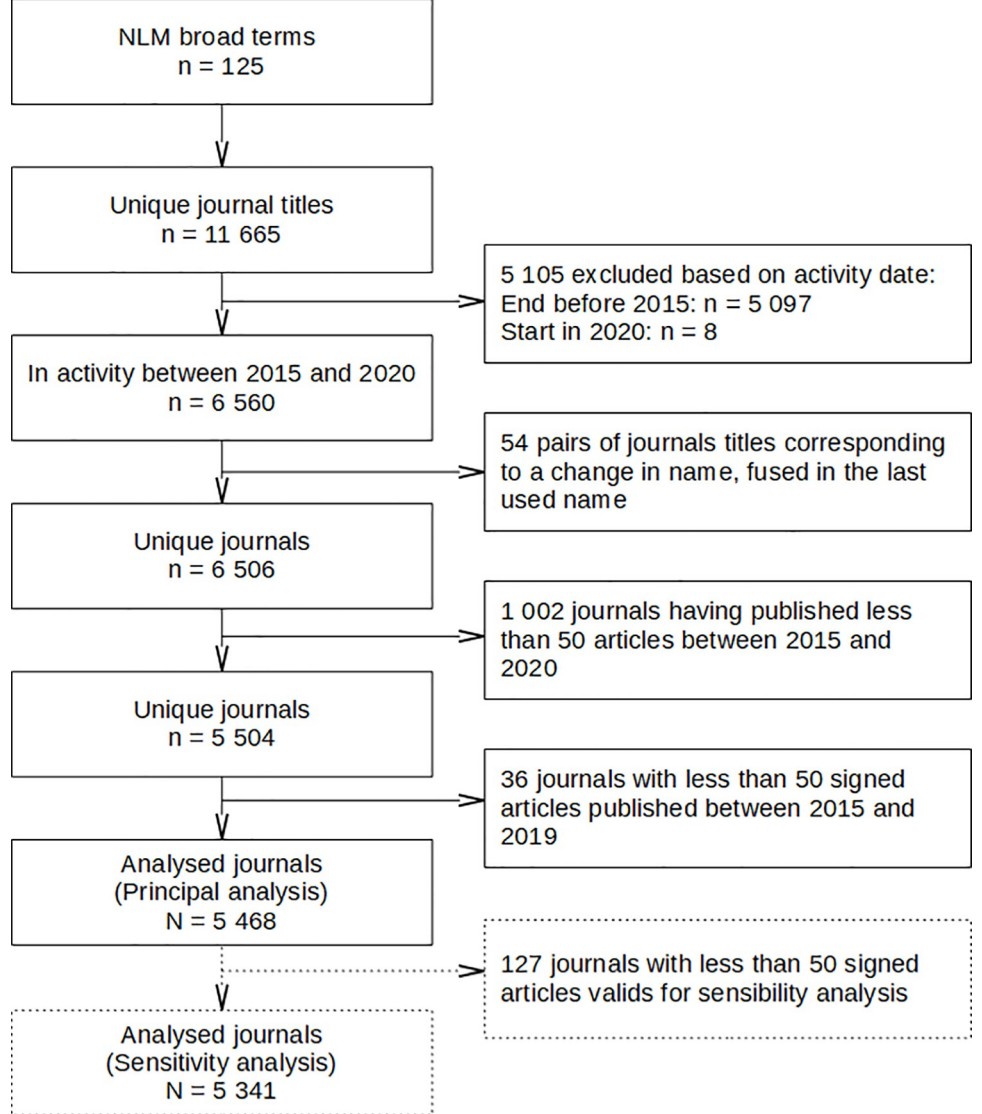

**Fig 1. Flow chart of included journals.** Selection flow chart for journals labeled with at least one "Broad Subject Term" in the NLM. NLM, National Library of Medicine.

details the reasons for noninclusion of some journals. After exclusions, 5,468 journals were analyzed in the principal analysis.

These journals published a total of 4,986,335 articles of which 4,183,917 were considered research articles (i.e., original articles, case reports, and reviews; **S1 Fig**). The main characteristics of the journals analyzed are described in **Table 1**. Briefly, they published a median of 500 articles (IQR 262 to 964) for the 2015 to 2019 period, of which 426 (IQR 232 to 798) were considered research articles. Two "mega-journals" published more than 25,000 articles over the 5-year period (Scientific Reports with 95,900 articles, and *PLOS ONE* with 108,990 articles). For 3,668 journals (67%), there was at least one article without any author, and a median percentage of 0.9% of articles (IQR 0.4% to 2.1%) with no named author in these journals. The author with the largest number of articles in a given journal is a journalist ($N = 767$), and the author with the largest number of research articles is an academic ($N = 252$). Both authors

**Table 1. Main characteristics of all journals in the United States NLM catalog having at least one Broad Subject Term and having published at least 50 authored articles between 2015 and 2019.**

| | | All articles | Research Articles |
|---|---|---|---|
| **Number of journals with ≥50 authored articles** | | 5,468 | 5,341 |
| **Number of articles** | | | |
| | Median [IQR] | 500 [262–964] | 426 [232–798] |
| | Range | 50–108,990 | 50–103,647 |
| **Number of articles with an author** | | | |
| | Median [IQR] | 494 [257–952] | 425 [232–795] |
| | Range | 50–107,342 | 50–103,647 |
| **PPMP (%)** | | | |
| | Median [IQR] | 2.88 [1.71–4.91] | 2.56 [1.61–4.21] |
| | Range | 0.1–39.9 | 0.126–44.8 |
| **PPMP 95th percentile (%)** | | 10.6 | 8.77 |
| **Number of articles by MPA** | | | |
| | Median [IQR] | 14 [8–25] | 11 [7–18] |
| | Range | 1–767 | 1–252 |
| **Tied as MPA** | | 1,022 (19%) | 1,271 (24%) |
| **Gini index** | | | |
| | Median [IQR] | 0.183 [0.131–0.246] | 0.161 [0.113–0.219] |
| | Range | 0.00–0.740 | 0.00–0.713 |
| **Gini index 95th percentile** | | 0.355 | 0.324 |
| **Median of publication lag ratio (MPA/no MPA)** | | | |
| | Median [IQR] | 0.829 [0.606–1.02] | 0.879 [0.697–1.04] |
| | Range | 0.00–26.8 | 0.0–754 |
| | Not calculable | 2,743 (50%) | 2,551 (48%) |

NLM, National Library of Medicine; MPA, Most Prolific Author; PPMP, Percentage of Papers by the Most Prolific author.

publish in established journals (The BMJ and Physical Review Letters, respectively) and are members of their respective editorial boards.

## Description of the indices

**Percentage of papers by the most prolific author and the Gini index.** For the principal analyses based on all articles published during the 2015 to 2019 period, the PPMP median was 2.88% (IQR 1.71% to 4.91%), and the 95th percentile of the PPMP value was 10.6%. **Fig 2A** details the PPMP and numbers of published outputs for all journals. It also shows that the number of publications by a prolific author in a given journal can result in different PPMP depending on the number of papers published in this journal. The most prolific author(s) in each journal published a median of 14 articles (IQR 8 to 25). For 1,022 journals (19%), there was more than one author with the same largest number of published articles. To assess authorship disparity from not only a single highly prolific author but from a group of authors, we also computed the Gini index, a measure of the degree of unequal distribution of author-ship within a journal (**S2 Fig**). Gini index range from 0 to 1, with smaller values indicating a more equal distribution of articles across authors and higher values representing greater inequality. Over the 2015 to 2019 period, the Gini index median was 0.183 (IQR 0.131 to 0.246), and the 95th percentile was 0.355. **Fig 2B** details the Gini indices and numbers of

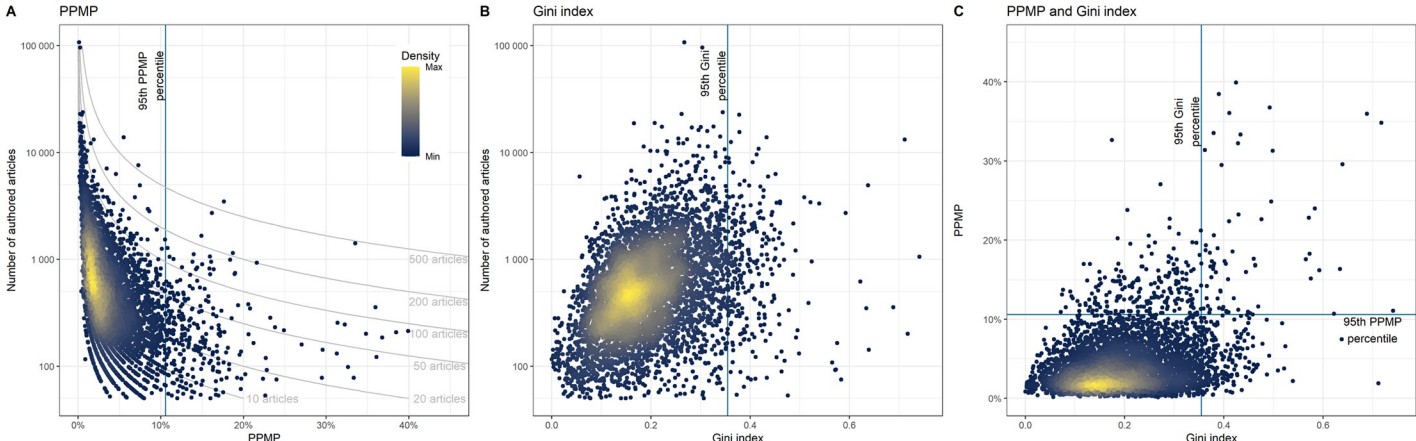

**Fig 2. PPMP and Gini index among all articles.** Distribution of PPMP author(s) (A) and Gini index (B) in relation to journal size, and comparison between the PPMP and the Gini index (C), across all articles published by all journals in the United States NLM catalog having at least one Broad Subject Term and having published at least 50 authored articles between 2015 and 2019. The data underlying this figure may be found in https://osf.io/6e3uf/. NLM, National Library of Medicine; PPMP, Percentage of Papers by the Most Prolific author.

published outputs for all journals. **Fig 2C** presents the correlation between the PPMP and Gini indices. This correlation was 0.35 (95% CI 0.33 to 0.37).

For both indices, there were no meaningful differences between index values across years, with the 95th percentile ranging from 10.1% to 11.4% for the PPMP and 0.212 to 0.224 for the Gini index (**S1 Table**).

Results of the sensitivity analyses based on "research articles" alone were consistent with those for all articles (**S3 Fig**). Correlations between indices computed for all articles and for "research articles" alone were 0.84 (0.84 to 0.85) for the PPMP and 0.93 (0.93 to 0.94) for the Gini index.

In 540 journals (9.9%), for at least a quarter of the authors, only the initials of their first name(s) were presented.

**Field-specific variations.** The distribution of the PPMP and Gini index for each NLM broad term is presented in **S4 and S5 Figs**. The median PPMP per field ranged from 1.1% to 9.5%, and the median Gini index per field ranged from 0.113 to 0.297.

**Publication lag.** Because of failures to report submission or acceptance dates, publication lag was not calculable for 2,743 journals (50.2%). Compared to journals that did report submission and acceptance dates, these journals had fewer authored articles (369 (IQR 200 to 712) versus 637 (IQR 355 to 1,186)). There were no differences for the Gini index but a higher PPMP (3.4% (IQR 2.0% to 5.9%) versus 2.4% (IQR 1.5% to 4.0%)). For the 2,725 journals with data on submission and publication (49.8%), the median of publication lag for all authored articles over the 5 years was 85 days (IQR 53 to 123) for articles published by the most prolific author(s) versus 107 days (IQR 80 to 141) for articles not published by the most prolific author (s).

**Fig 3** shows the scatter plot for all articles with the marginal density curve for the median of publication lag for the most prolific author(s) versus nonprolific authors, for each journal: For articles authored by the most prolific author(s), the distribution of the publication lag was skewed toward a shorter time lag. Using a cutoff of 3 weeks for the median of publication lag, 277 (10.2%) of the journals had a median below this for articles by the most prolific author(s), 51 (1.9%) journals had a median below this for articles not by prolific author(s), and 38 (1.4%) journals had a median below this for both types of articles (i.e., authored by the most prolific

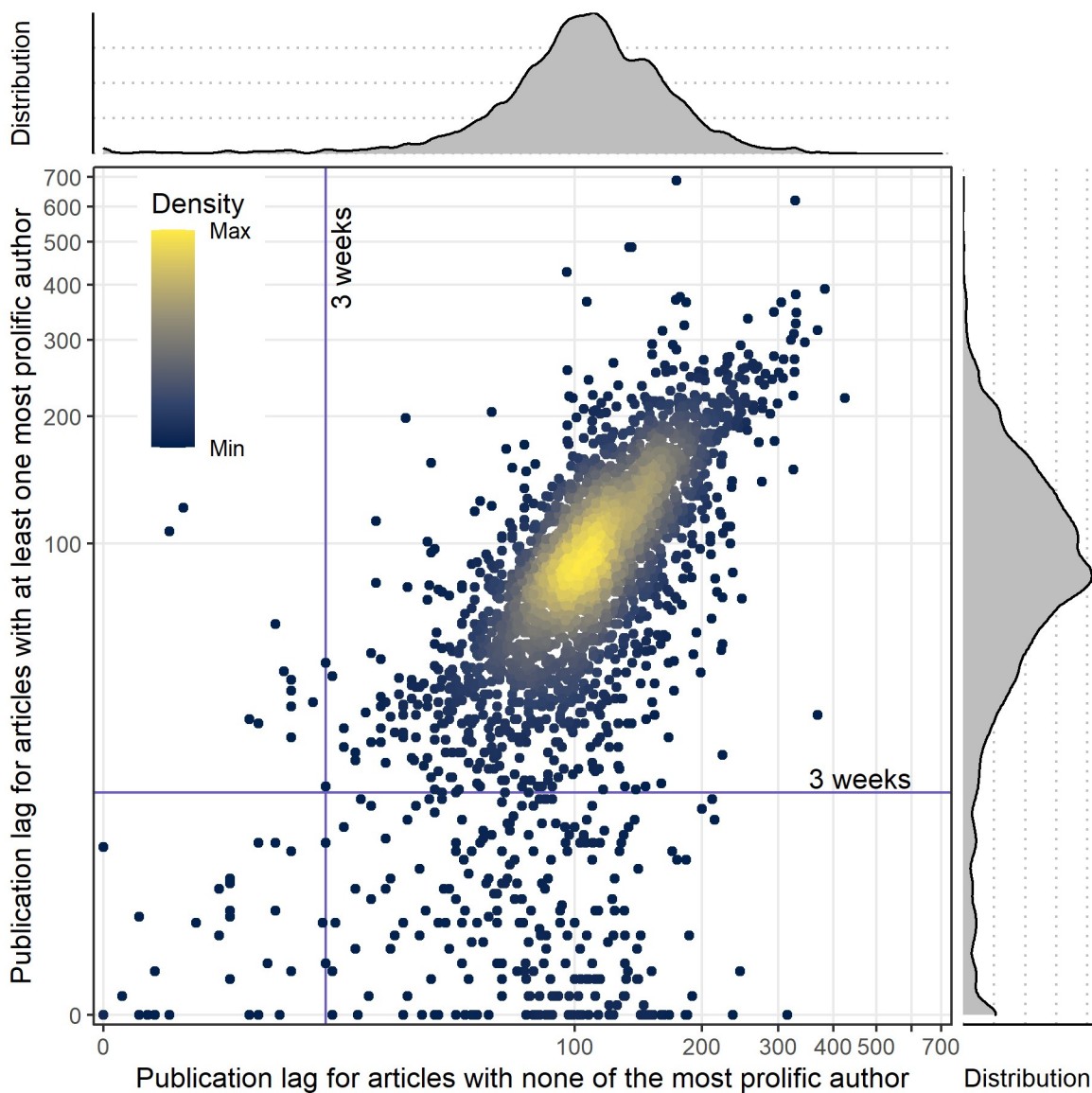

**Fig 3. Publication lag.** Distribution of the publication lag median for the subgroup of 2,725 (49.8%) journals reporting submission and publication dates. Publication lag median (in days) are presented for articles signed by the most prolific authors compared to the articles without any of the most prolific authors (with marginal density plot of distributions). The data underlying this figure may be found in https://osf.io/6e3uf/.

author(s) or not). For the most prolific authors, publication lag decreased with the number of articles published as shown in **Fig 4**, not solely in outlier journals. The results of the sensitivity analyses based on "research articles" alone were consistent with those for all articles (**S6 and S7 Figs**).

## Description of outliers and identification of nepotistic journals

Using the 95th percentile value, we identified through the principal analysis 480 outlier journals: 206 based on the PPMP and the Gini index considered separately, and 68 based on both

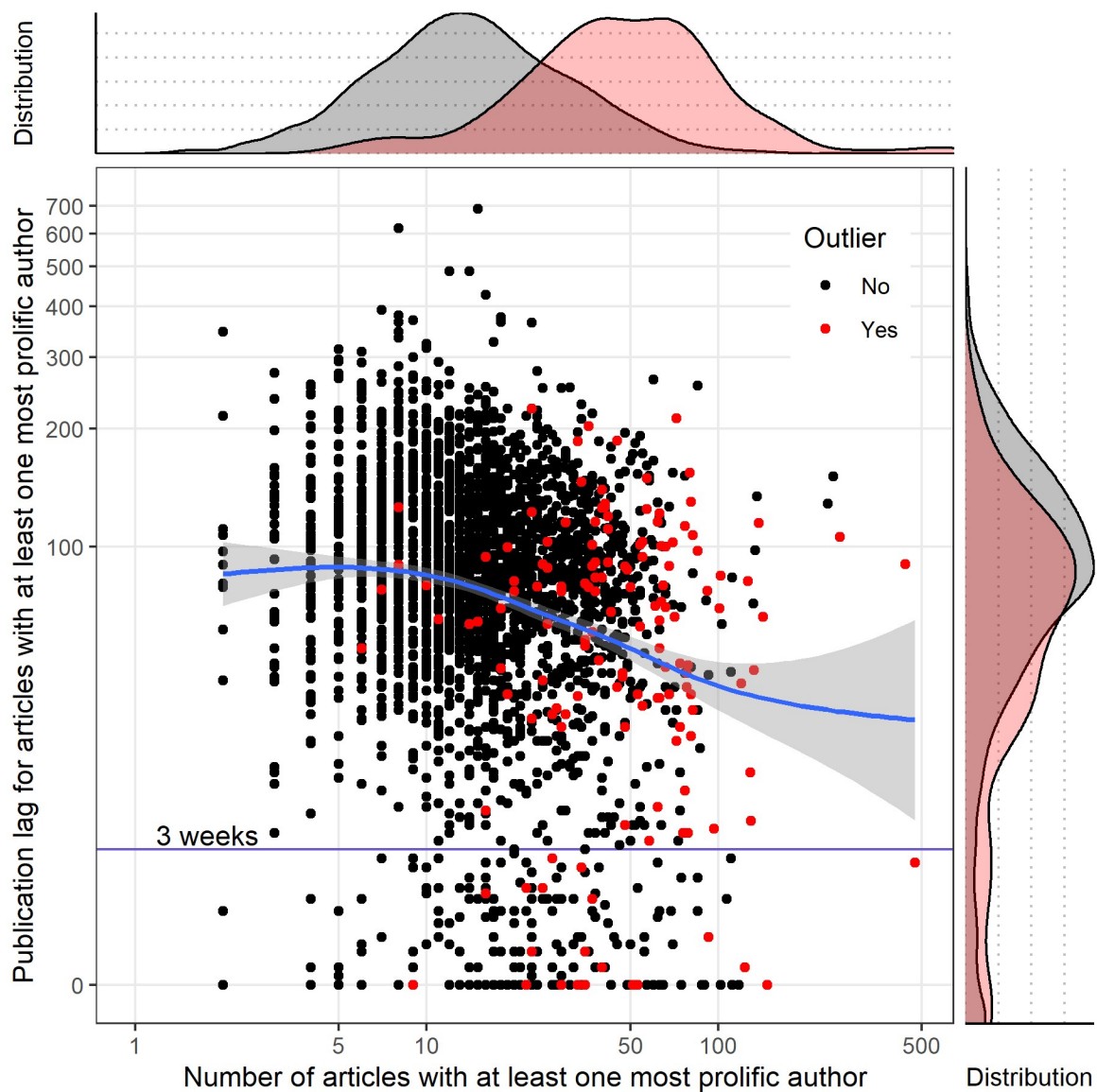

**Fig 4. Publication lag among articles of the most prolific author.** Distribution of publication lag (in days) and number of articles authored for each of the most prolific authors, across all articles (with marginal density plot of distributions) for the subgroup of 2,725 (49.8%) journals reporting submission and publication dates. The data underlying this figure may be found in https://osf.io/6e3uf/.

indices. The yearly and global distributions of these indices are presented for all journal in **S2 Table**.

The main characteristics of the 100 randomly selected outlier journals are presented in **S3 Table**. Of the 100 journals identified through the principal analyses, 98 were reported in English, among which 31 were also in another language (either fully multilingual journals or translation of abstracts). The most common non-English languages were German (6 journals), Chinese (5 journals), Japanese (5 journals), and French and Italian (4 journals each). These outliers were well-established journals, with a median year of start of activity in 1990 (IQR 1976 to 2001).

Only 56 of these 100 journals were indexed in Web of Science (WoS), which enables an assessment of the journal impact factor and other citation metrics. For these 56, the median journal impact factor was 2.9 (IQR 1.5 to 4.8) with a median self-citation ratio of 0.11 (IQR 0.047 to 0.21), corresponding to a median self-citing boost according to Ioannidis and Thombs of 13% (IQR 5% to 26%) [6]. The skewness and nonarticle inflation median was 86% (IQR 55% to 144%) [6]. Calculation of this metric was not possible for 11 journals (20%) that had a median of article citation of 0. Only 5 journals were indexed in the Directory of Open Access Journals as being full open access, and the median proportion of open access articles was 2.0% (IQR 0.47% to 8.0%). None of the outliers had an open peer review policy.

For 2 of the 100 journals, the full composition of the editorial boards could not be found and only the editor-in-chief was known, but was not the most prolific author. In the remaining 98 journals, at least one of the most prolific authors was a member of the editorial board in 60 journals (61%), among whom 25 (26%) were the editors-in-chief. Journals where at least one of the most prolific authors was on the editorial board tended to have a higher impact factor than the others with a median of 3.4 (IQR 2.0 to 5.3) versus a median of 1.4 (IQR 1.0 to 3.1).

Because there was sometimes more than one author with the same largest number of published articles, 108 "most prolific" authors were identified in the 100 outlier journals. We identified errors in author identification for one journal, MMW-Fortschritte der Medizin (an outlier on the Gini index), where the most prolific "author" was named "Red," which seems to be a diminutive for "Redaktion," possibly encompassing several physical individuals. When "Red" was ruled out as a valid author name (i.e., these papers were considered has having no author), the next most prolific author was, however, a member of the editorial board, the PPMP increased very slightly from 7.5% of 4,920 authored articles to 7.6% of 4,553 authored articles, and the Gini index decreased very slightly from 0.637 to 0.616. Out of 1,978 identified authors (excepting "Red"), 1,435 (72.5%) did not publish more than one article in this journal. Among the remaining 107 individual authors, 95 were formally identified from WoS. Among the 12 remaining, identification was considered unreliable for 8 because of possible homonyms, and 4 were not indexed in WoS. For these 12 authors, manual disambiguation using PubMed and Google identified 6 journalists, 5 physicians with a consistent affiliation to the journal considered, and one author where a high risk of homonym persisted ("Wang, Y" in Journal of Clinical Otorhinolaryngology, Head, and Neck Surgery). Among the 95 other authors, the median of the H-index was 28 (IQR 13 to 50).

Overlap between indices (PPMP versus Gini index) and analyses (principal versus sensitivity analyses) is presented in **S8 Fig**. Among the 648 journals flagged as outliers either through the principal analyses, or through the sensitivity analyses, 299 (46.1%) are common to both analyses. Furthermore, qualitative analysis of the 100 randomly selected outlier journals identified through the sensitivity analyses is consistent with those for the principal analyses (see **S3 Table**).

## Discussion

In this comprehensive survey of 5,468 biomedical journals, we describe several features of editor–author relationships among which were the following: (i) an article output was sometimes dominated by the prolific contribution of one author or a group of authors; (ii) with time lags to publication that were in some instances shorter for these prolific authors (when this information was available); and (iii) more than half of those prolific authors were typically members of the journal's editorial board.

In our study, the threshold to define the 95th percentile was 10.6% of articles published by the most prolific author. In absolute terms, we believe that it is reasonable to closely inspect

these journals as it may question the judgment of an editor where more than 10% of the published papers are authored by the same person. To better characterize the lack of heterogeneity in authorship, we also computed the Gini index, for which the corresponding 95th percentile was 0.355 over 5 years and 0.20 when computed annually. One possible explanation is that a broader time frame allowed for more occasional authors to be recruited while maintaining the regular authors, revealing the latent heterogeneity. One other explanation is that some recent journals are relying on a group of authors on their first year before allowing these authors to move on to other publishing duties.

The PPMP and the Gini index explore complementary patterns of asymmetry in publishing patterns. The PPMP reflects author practices, while the Gini index is more sensitive to groups of highly prolific authors. The Gini index has one advantage over the PPMP, in that it is less constrained by the total number of articles published. If a journal publishes a very large number of papers (i.e., 1,000 articles over 12 months), it becomes increasingly implausible that a single prolific author could account for 10% or more of them, as there is a natural upper limit to how many papers any one individual can author. Conversely, for a journal that publishes very few papers, an author could be identified as above the 95th percentile with a relatively modest number of publications. In other words, there might be a masking effect in journals with higher rates of publishing causing lower publishing rate journals to show a greater number of PPMPs. Further developments may consider breaking the journals into groups based on rates of publishing and/or using crude number of publications by the most prolific author.

For the subgroup of 2,725 (49.8%) journals reporting submission and publication dates, the time lag for the most prolific author(s) was shorter, suggesting that for certain journals, which are outliers on PPMP and/or Gini index, peer reviews may have been absent or only superficial for prolific authors. However, for the most prolific authors, the publication lag decreases with the number of articles published across all journals and not solely in the subsample of outliers. This suggests that our description of these outliers based on PPMP could be only the tip of the iceberg, capturing solely the most extreme cases of hyperprolific publication in a given journal. Because not all journals report publication lag data, this finding is necessarily based on a subgroup. This could introduce bias if journals reporting publication dates differ from those that do not in terms of factors such as size or monitoring of the publication process [7].

Our findings persisted when all articles were considered as well as when only "research articles" were considered (i.e., excluding articles explicitly referenced as editorial, correspondence, or news articles), suggesting that editorials, correspondence, and news are not the only drivers of the indicators we explored. Conversely, it is possible that the definition we used to identify research articles is not perfect and carries a risk of misclassification bias. However, analyzing both the overlap of PPMP and Gini and the overlap between both analysis populations cautions against making simple binary distinctions. When based on all articles, these metrics carry a risk of false positives, represented by active editors and/or professional writers. When based on research articles, it may miss problematic behaviors by certain editors. In other words, there is surely a gray zone; the proposed metrics are useful to delineate the big picture of medical journals behaviors, but the specific behaviors of these journals will necessarily deserve more fine-grained scrutiny in future. For instance, it will be interesting to see whether hyperprolific authors publishing in their own journals adhere to COPE policy regarding conflicts of interest disclosures [8]. This was studied for authors of Cochrane reviews who are also editorial board members: The adherence to conflicts of interest policy was low [9].

We should beware of assuming that a hyperprolific author is necessarily engaged in questionable publishing practices: Some people are highly productive, and the speed with which good research can be completed is highly variable across research fields. Furthermore, authors may be represented in many papers because they play a key role in one aspect of the research,

such as the statistical analysis, and senior researchers who oversee multiple projects may end up as authors on many papers. Similarly, shorter publication lags may occur simply because it is easier to find reviewers for eminent authors, or in a particular subject area, and/or because their expertise means that their papers require less revision. Nevertheless, there is no doubt that some highly prolific authors achieve an unusual level of productivity by exploiting the system or engaging in academic misconduct [10,11]. It is important to make a distinction between hyperprolific authors who publish a lot in a range of different journals and those who are exploiting a select pool of a few journals in which they appear as prominent authors (as we explored here). It is also very important to complement the PPMP and the Gini index with the absolute numbers of papers authored by the most prolific authors, because some problematic journal behaviors could pass unnoticed when only these 2 indices are used.

On a random sample of outlier journals identified using the PPMP and/or the Gini index, we found that the prolific authors can be "established" scientists with a relatively high H-Index (for instance, a median of 28), and that 60% of these most prolific authors were editors-in-chief or members of the journal's editorial board. About half of these journals had a median journal impact factor of 2.9 (IQR 1.5 to 4.8). These journals generally presented a large self-citation ratio, meaning that some of them may have questionable practices by manipulating their impact factor. The other half of these journals did not have an impact factor, possibly indicating they were new journals joining the WoS. Even though WoS uses an extensive list of eligibility criteria, it is also possible that some of the new journals are predatory, journals that are known to have "leaked into" trusted sources [7]. Importantly, while NLM is skewed toward English language journals, several outlier journals were in other languages. These journals are likely to represent smaller communities with the possibility of closer interactions between researchers.

Our results underscore possible problematic relationships between authors who sit on editorial boards and decision-making editors. Typically, publishers promote independence between authors and journals. Hyperpublished authors may see such relationships as a way to more easily reach publication thresholds for hiring, promotion, and tenure. There may be defensible reasons for members of the editorial board of a journal to hyperpublish in a journal [3]. There are, for instance, certain research fields that are research niches, where the contributing authors are part of a very small community of specialists and are therefore the most likely authors.

Although our findings are based solely on a subsample of journals, they provide crucial evidence that editorial decisions were not only unusually, but also selectively, fast for the favored subset of prolific authors. This pattern was also found by Sarigöl and colleagues when exploring favoritism toward collaborators and coeditors, which persists even after taking into account individual article quality, measured as citation and download numbers [7]. This phenomenon could have an impact on productivity-based metrics and suggests a risk of instrumentalization, if not corruption, of the scientific enterprise, by using journals as a "publication laundry" for "vanity publication" [12] by authors closely related to the editorial board. Exploiting productivity metrics has been widely described, in the form of self-citation, honorary authorship, and/or ghost writing. Manipulation of individual metrics by resorting to a dedicated "nepotistic" journal appears to be a little studied way of exploiting the system.

## Limitations

Our descriptive and exploratory survey, based on a large available database, provides information about the broad scene of "nepotistic journals," but it may miss some finer points, especially concerning the quality of articles published in these journals. The quality of a scientific

article is a difficult concept to measure, and it cannot be easily summarized in quantitative metrics. We recommend a qualitative analysis of the papers published by the most prolific authors in journals flagged by these indicators, as well as an analysis of the relationship between this author and the editorial board. In addition, we restricted our analysis to journals indexed in the NLM under one or more of the existing broad terms. Some journals are registered without broad terms, requiring a manual pickup by the NLM. Consequently, our survey may have preferentially included the more established journals indexed in the NLM, with a durable presence in the database, and hence likely to have a better-quality global and editorial conduct than nonindexed journals. Similarly, because we restricted our search to journals publishing a minimum of 50 papers in the 2015 to 2019 period, we may have missed smaller journals with less professional editorial staff and/or from publishers with different standards.

Importantly, our automated calculations rely on the articles identified through an NLM search, and it is possible that not all journals list all articles within PubMed. Although this is likely rare, we cannot exclude that certain authors use somewhat different name (for instance, change of name after marriage). We could not explore this bias that would have resulted in an underestimation of our indicators. More importantly, these calculations carry a risk of inaccuracy as a result of homonymy. Misidentification and/or merging of author names could bias the PPMP and the Gini index in both directions, and the risk of merging increases when only the initial or first name is known, and in the case of authors with similar names. The greater risk of homonyms could partially explain the increased Gini index values for larger journals, without reference to a tendency to editorial misconduct. Our analysis of the random sample of outliers enabled a disambiguation procedure consisting in inspecting qualitatively the most prolific authors. Only 1 out of 108 "most prolific" authors within a given journal was considered to be at being at high risk of homonymy. Among these 108 authors, this procedure also enabled identification of the 6 most prolific authors, who were professional journalists for whom high productivity is of course not an indicator of any academic misconduct, as they are professionals paid by the journal and not academics. The 2 proposed indicators, and their current calculation, should therefore not be used indiscriminately but could rather serve as a screening tool for potentially problematic journals that may then require careful exploration of their editorial practices. In addition, by analogy with citation-based metrics [13], we believe that no single metric can be sufficient but rather that different metrics can be complementary to inform about editorial behavior and that these metrics must not be used indiscriminately without considering all the identified limitations.

While our results are exploratory and do not yet support a widespread use of these indicators, we hope that further research will help to establish these easily computed indexes as a resource for publishers, authors, and indeed scientific committees involved in promotion and tenure, to screen for potentially biased journals needing further investigation. DORA paved the way, of moving away from productivity-based metrics, and other efforts followed such as the Hong Kong Principles for assessing researchers. Integrity-based metrics are indeed needed to overcome the limitations of productivity-based metrics [14]. A transparent declaration of interests in communicating research is surely one important aspect of scientific integrity and trustworthy science. This principle of course applies to financial conflicts of interest, which are often underdeclared by journal editors [15], and also to nonfinancial conflicts of interest such as editor–author relationships.

The proposed indices could add transparency in the editorial decision-making and peer review process of any journal. This transparency is currently lacking toward the public and any stakeholder involved in the research community, such as COPE, the Committee for Publication Ethics. Guidance for editors and publishers should be developed to delineate good practices and prevent obvious misconduct.

## Methods

We developed and followed a research protocol, which was prospectively registered on July 21, 2020, on the Open Science Framework (https://osf.io/6e3uf/). The analytic code and summarized data are also available on the same URL.

### Data extraction

The eligibility criteria for the selection of journals were the following: (i) a biomedical journal referenced in the NLM in the MEDLINE database; (ii) having at least one "Broad Subject Term"; and (iii) having published more than 50 papers between January 2015 and December 2019. "Broad Subject Terms" are Medical Subject Headings (MeSH), terms used to describe a journal's overall scope, and they are defined by the NLM for journals in the MEDLINE database [16]. Each journal was analyzed only once regardless of the number of "Broad Subject Term" associated with the journal (except in subgroup analysis by "Broad Subject Terms"). The 2015 to 2019 period was chosen, as this 5-year window enables a smoothing of random variations and description of recent practices. One author (AS) searched for changes to journal names during the 2015 to 2019 period and, in cases of renaming, pooled the articles published under the different names.

To identify eligible journals, we used the Entrez programming utilities (E-utilities), which enable queries to the National Center for Biotechnology Information (NCBI) databases. The search query—presented in **S1 Text**—was used to identify all biomedical journals in the NLM catalog having at least one of the "Broad Subject Terms" listed. Then, for each journal, article metadata was automatically collected with E-utilities. On account of technical restrictions, querying for article metadata was run from 2015 up to the date of extraction and then restricted to the period January 2015 to December 2019.

To manage articles without an author name, the third selection criterion was slightly modified to focus on journals with at least 50 "authored articles"—i.e., articles with at least one identified author—over the 2015 to 2019 period (see "Protocol changes"). Publications reprinted in several journals (for instance, PRISMA statements published in 6 different journals to promote dissemination) did not receive special treatment, and no correction was applied, as each article was only examined in relation to its publication journal.

### Index calculation

**Percentage of papers by the most prolific author and Gini index.** For each journal, each author was identified by his or her full name (i.e., family name and complete first name) or barring that, by his or her family name and first name initial(s). The number of articles authored by this person was counted. When there was more than one author with the same largest number of published articles, they were all considered as the "most prolific" authors. The PPMP was defined as the number of articles by the most prolific author ($n_{max}$) divided by the total number of authored articles in the journal ($N_{tot}$): PPMP = [$n_{max}$ / $N_{tot}$]. Complementary to the PPMP, the Gini index was used to explore inequality in the number of published articles related to more than one author. The Gini index for the number of publications by each author was calculated, with correction for the total number of authors (see formula and example in **S2 Fig**) [4]. Gini index range from 0 to 1, with smaller values indicates a more equal distribution of articles across authors and higher values represent greater inequality.

For the primary analysis, these 2 outcomes (PPMP and Gini index) were computed for all papers (including research articles, editorials, comments, etc.), and for a sensitivity analysis, they were computed only for research articles (using the NCBI publication type). In line with previous works [5,17], articles that were considered as research articles were included if (i) the

**Publication Type** field was coded "Journal Article" and if (ii) the **Abstract** field was not empty. Furthermore, articles were not considered as research articles if **Publication Type** field was coded with the following label: "Comment," "Letter," "Editorial," "Published Erratum," "News," "Introductory Journal Article," "Biography," "Portrait," "Congress," "Interview," "Retraction of Publication," "Personal Narrative," "Retracted Publication," "Patient Education Handout," "Lecture," "Autobiography," "Clinical Conference," "Classical Article," "Address," "Legal Case," "Expression of Concern," "Festschrift," "Overall," "Bibliography," "Corrected and Republished Article," "Interactive Tutorial," "Duplicate Publication," "Directory," "Newspaper Article," "Periodical Index," "Dictionary."

**Publication time lag.**   For each article, the publication lag—defined as the time between submission and acceptance of an article—was computed whenever possible. After this, each journal was characterized by (i) median publication lag for articles authored by at least one of the most prolific authors and (ii) median publication lag for articles not authored by the most prolific author(s).

## Description of outlier journals

Outliers were defined as journals with a PPMP value and/or the Gini index above their respective 95th percentiles in the principal analysis (i.e., on all articles) and in the sensitivity analysis (i.e., on research articles). For pragmatic reasons, 2 samples of 100 outlier journals were randomly selected (first sorted by full name, in alphabetic order, and randomly sampled using a random number generator with a seed arbitrarily set at 42; R function sample_n in dplyr package).

One reviewer (AS or FN or CL) manually extracted characteristics related to the journal impact factor (WoS), open access policies (WoS and Directory of Open Access Journals), open peer review policies (Publons–Clarivate), the most prolific authors' H-index (WoS), and presence and role (i.e., editor-in-chief or board member) on the editorial board of the journal (journal or publisher website). Where this information was available, we made a distinction between advisory boards (that were not considered in the analysis) and editorial boards. For the year 2019, the metrics "self-citation boost" (i.e., number of self-citing articles over number of non-self-citing articles) and "skewness and nonarticle inflation" (i.e., impact factor minus median of citations for an article, over median of citations for an article) was computed according to Ioannidis and Thombs [6]. Importantly, this extraction allowed for qualitatively exploring the possibility of homonyms between authors by comparing names, affiliations, research field, and all available qualitative information on NLM and WoS (using WoS Author search tool). When a doubt persisted, we used Google to identify the author at risk of homonymy and their credentials.

## Data analysis

A descriptive analysis was performed using median, range, and quartiles for continuous variables, and counts and percentages for categorical variables. For both analyses, descriptions for the 100 outlier journals were computed overall and with respect to membership of any of the most prolific author(s) on the editorial board. Correlations were computed using Pearson's coefficient, with 95% confidence interval (CI).

To explore field-specific variations, the distribution of the 2 indices within each "Broad Subject Term" was graphically displayed. The yearly and overall distribution of the percentage of papers by each author and the Lorenz curve—a graphic representation of the cumulative distribution of appearances as an author—were presented for each of the potential outliers identified above.

All analyses were conducted using R version 3.6 [18], and main packages RISmed 2.1 for queries on journal characteristics [19], easyPubMed 2.13 for queries on article characteristics [20], DescTools 0.99 for Gini index calculation [21], and tidyverse 1.3 for miscellaneous [22].

### Protocol changes

Some practical unforeseen challenges arose in our research because a few articles unexpectedly lacked author names (i.e., articles without authors), which precluded them from contributing to the numerator of PPMP or to the Gini index. We therefore amended our definitions to make it explicit that the PPMP denominator was defined as the number of articles with at least one identified author rather than all published articles, and journals were included only if they had published 50 articles with author names rather than all published articles.

The 3-week threshold used to describe publication lag as being suggestive of unduly rapid or absent peer review was not initially specified in our protocol and was arbitrarily added for descriptive purposes. Three weeks seems plausible for a thorough review process while expedited reviews in a few days can be suspicious if there is a pattern of fast reviews—and especially if those fast reviews applied selectively to certain favored authors). We also explored the relationship between publication lag for articles authored by any of the most prolific author(s) and the number of papers authored by these authors. The description of the outlier journals with respect to the membership of any of the most prolific author(s) on the editorial board was added a posteriori for exploratory purposes.

We initially planned to focus our sensitivity analysis on "journal articles" only. During the peer review process, following a Science's new [23], it appeared clear that this category was not specific enough. The protocol was therefore edited (https://osf.io/6evmz/) with a better definition of "research articles." We have also described the overlap between both analysis and added more emphasis on the sensitivity analysis by describing a random sample of outliers in this analysis (it was not part of our initial protocol).

### Supporting information

**S1 Fig. Cross classification of Publication Types.** Publication types extracted from MEDLINE Metadata to each article and their co-occurrences, among all journals in the United States NLM catalog having at least one Broad Subject term and having published at least 50 signed articles between 2015 and 2019. The data underlying this figure may be found in https://osf.io/6e3uf/. NLM, National Library of Medicine.
(TIF)

**S2 Fig. Formula and example to understand the Gini index.** We investigated the level of inequality in the distribution of authorship among authors using the Gini index. This statistical measure is derived from the Lorenz curve and is widely used in econometrics to describe income or wealth inequalities in a given population. In our study, "income" corresponds to the number of articles signed by authors in a given journal, and the "population" is all authors who have published at least one article between 2015 and 2019 in the journal. The Lorenz curve is a graphical representation of the ranked distribution of the cumulative percentage of authors on the abscissa versus the cumulative percentage of authorship distributed along the ordinate axis. In case of complete equality across author (i.e., each author within a journal has published the exact same number of articles), the Lorenz curve would follow the 45 degree diagonal. The further inequality increases (i.e., one author or a group of authors published more articles than others authors), the further the Lorenz curve moves away from this diagonal of equal distribution. The Gini index is a measure of the area between the diagonal of equal

distribution and the Lorenz curve, corrected for the number of authors $\frac{n}{n-1}$. In practice, the calculation formula is $\left[\frac{2\sum iy_i}{n\sum y_i} - \frac{n+1}{n}\right]\frac{n}{n-1}$, where n is the total number of authors, and $y_i$ is the number of articles published by author i, with authors sorted in nondecreasing order of article numbers. The Gini index ranges from 0 to 1, with smaller values indicating a more equal distribution of articles across authors and higher values representing greater inequality. As expressed in its formula, the Gini index calculation gives a higher weight to extreme positive values and may be comparable in interpretation to a normalized root-mean-square error against an expected distribution of "all authors appear exactly the same number of times." As a toy example, hypothetical cases of distribution of authorship between 3 authors totaling 24 authorships (left), with corresponding Lorenz curve and Gini index (right) are shown in the figure below. The dark blue line represents the diagonal of equal distribution (scenario 1), and other lines represents Lorenz curve of different scenarios of inequality (scenarios 2–5). The Gini formula counts the times an author's name occurs but does not distinguish between papers contributed by different authors. Thus, the scenario 1 shown here could occur with 8 articles published in common by the 3 authors, or 4 articles published in common and each of the 3 separately publishing 4 other articles.
(TIF)

**S3 Fig. PPMP and Gini index among articles considered as research articles.** Distribution of the (A) PPMP author(s) and (B) the Gini index in relation to journal size, and (C) comparison between the PPMP and the Gini index, among articles considered as research articles (i.e., original article, case reports, and reviews), published by all journals in the US NLM catalog having at least one Broad Subject term and having published at least 50 signed articles between 2015 and 2019. The data underlying this figure may be found in https://osf.io/6e3uf/. NLM, National Library of Medicine; PPMP, Percentage of Papers by the Most Prolific author.
(TIF)

**S4 Fig. PPMP according to the subject area.** Distribution of the PPMP author for each US NLM broad term represented by at least 10 journals having published at least 50 signed articles between 2015 and 2019. The number of journals covered by a Broad Subject term is shown next to the name of the field of study. Width of the box-and-whisker relative to the number of journals. Vertical line at the 95th percentile of PPMP among journals. The data underlying this figure may be found in https://osf.io/6e3uf/. NLM, National Library of Medicine; PPMP, Percentage of Papers by the Most Prolific author.
(TIF)

**S5 Fig. Gini index according to the subject area.** Distribution of the Gini index for each US NLM broad term represented by at least 10 journals having published at least 50 signed articles between 2015 and 2019. The number of journals covered by a Broad Subject term is shown next to the name of the field of study. Width of the box-and-whisker relative to the number of journals. Vertical line at the 95th percentile of PPMP among journals. The data underlying this figure may be found in https://osf.io/6e3uf/. NLM, National Library of Medicine; PPMP, Percentage of Papers by the Most Prolific author.
(TIF)

**S6 Fig. Publication lag for articles considered as research articles.** Distribution of the publication lag median (with marginal density plot of distributions) for the subgroup of 2,790 (52.2%) journals reporting submission and publication dates. Publication lag median (in days) are presented for articles signed by the most prolific authors compared to the articles without any of the most prolific authors, among articles considered as research articles (i.e., original

article, case reports, and reviews). The data underlying this figure may be found in https://osf.io/6e3uf/.
(TIF)

**S7 Fig. Publication lag for articles considered as research articles with at least one most prolific author.** Distribution of publication lag median (with marginal density plot of distributions) and number of articles authored for each of the most prolific authors, across articles considered as research articles (i.e., original article, case reports, and reviews) for the subgroup of 2,790 (52.2%) journals reporting submission and publication dates. The data underlying this figure may be found in https://osf.io/6e3uf/.
(TIF)

**S8 Fig. Venn diagram illustrating overlap between indices (PPMP versus Gini index) and analyses (principal versus sensitivity analyses).** The data underlying this figure may be found in https://osf.io/6e3uf/. PPMP, Percentage of Papers by the Most Prolific author.
(TIF)

**S1 Table. Description of included journals.** Main characteristics of all journals in the US NLM catalog having at least one Broad Subject term and having published at least 50 signed articles between 2015 and 2019. NLM, National Library of Medicine.
(XLSX)

**S2 Table. Yearly and global individual data for the main characteristics of each selected journal ($N$ = 5,468).**
(XLSX)

**S3 Table. Main characteristics of 100 randomly selected outlier journals identified through the principal analyses and through the sensitivity analyses.** Journals are considered as outliers if they have a PPMP or a Gini index higher than the 95th percentile, among the journals in the US NLM catalog having at least one Broad Subject term and having published at least 50 signed articles between 2015 and 2019. NLM, National Library of Medicine.
(XLSX)

**S1 Text. US NLM catalog journal identification query.** NLM, National Library of Medicine.
(DOCX)

## Acknowledgments

We thank Angela Swaine Verdier for revising the English and Xavier Chard-Hutchinson, teaching librarian, for his assistance on NLM resources. Our thanks also go to Mathilde Calais and Alice Vinatier, who contributed to the extraction of the qualitative data for the sample of outliers in the sensitivity analysis.

## Author Contributions

**Conceptualization:** Alexandre Scanff, Florian Naudet, Ioana A. Cristea, David Moher, Dorothy V. M. Bishop, Clara Locher.

**Formal analysis:** Alexandre Scanff.

**Methodology:** Alexandre Scanff, Florian Naudet, Ioana A. Cristea, David Moher, Dorothy V. M. Bishop, Clara Locher.

**Project administration:** Florian Naudet, Clara Locher.

**Resources:** Alexandre Scanff.

**Software:** Alexandre Scanff.

**Supervision:** Clara Locher.

**Validation:** Alexandre Scanff, Clara Locher.

**Visualization:** Alexandre Scanff.

**Writing – original draft:** Florian Naudet, Dorothy V. M. Bishop, Clara Locher.

**Writing – review & editing:** Alexandre Scanff, Florian Naudet, Ioana A. Cristea, David Moher, Dorothy V. M. Bishop, Clara Locher.

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
