## [Editor Report · Decision Letter 0]

27 Jan 2021

Dear Dr Locher, 

Thank you for submitting your manuscript entitled "‘Nepotistic journals’: a survey of biomedical journals." for consideration as a Research Article by PLOS Biology.

Your manuscript has now been evaluated by the PLOS Biology editorial staff, as well as by an academic editor with relevant expertise, and I'm writing to let you know that we would like to send your submission out for external peer review.

IMPORTANT: We will be reviewing your manuscript as a Meta-Research Article. Please could you change the article type to "Meta-Research Article" when you upload your additional metadata (see next paragraph)? No re-formatting is required.

Please re-submit your manuscript within two working days, i.e. by Jan 29 2021 11:59PM.

Kind regards,

Roli Roberts

Senior Editor

PLOS Biology

---

## [Decision Letter · Decision Letter 1]

26 Mar 2021

Dear Dr Locher,

Thank you very much for submitting your manuscript "‘Nepotistic journals’: a survey of biomedical journals." for consideration as a Meta-Research Article at PLOS Biology. Your manuscript has been evaluated by the PLOS Biology editors, an Academic Editor with relevant expertise, and by three independent reviewers; in addition the Academic Editor has kindly provided some extra guidance that I have included in the foot of this email.

You'll see that the reviewers are broadly positive about you study, but each raises a number of concerns that must be addressed (especially rev #2, who has provided a very thorough set of comments). You should also attend to the additional comments provided by the Academic Editor (note that s/he considers one of the requests from reviewer #3 to be optional).

In light of the reviews (below), we will not be able to accept the current version of the manuscript, but we would welcome re-submission of a much-revised version that takes into account the reviewers' comments and those of the Academic Editor. We cannot make any decision about publication until we have seen the revised manuscript and your response to the reviewers' comments. Your revised manuscript is also likely to be sent for further evaluation by the reviewers.

We expect to receive your revised manuscript within 3 months. 

**IMPORTANT - SUBMITTING YOUR REVISION**

*Re-submission Checklist*

*Published Peer Review*

*PLOS Data Policy*

*Blot and Gel Data Policy*

Sincerely,

Roli Roberts

Senior Editor,

rroberts@plos.org,

PLOS Biology

REVIEWERS' COMMENTS:

Reviewer #1:

The authors perform an analysis related to research integrity, in particular of journals and their editors. For this they use the percentage of papers by the most prolifc author (really a red flag if too high) and the Gini index of contributions over all authors of the journal ( a more refined indicator in this case). They further used the median lag time to publication, especially for the most prolific author (another red flag to signal journals with questionable behavior). As a bibliometrician I testify that the authors used the appropriate methods. Of course, the authors could have used another concentration index, but I see no reason why they should in this case. 

I have two minor observations:

1) I do not understand the meaning in Fig. 4 of "number of articles with any most prolific author" . Do they mean "at least one" ? 

2) the box and whisker graphs pages 26/27 are extremely small. Maybe they can be included, in a larger version, as supplementary material.

Reviewer #2:

[identify themselves as Ivan Oransky and Alison Abritis]

Thank you for the opportunity to review this manuscript, which is important as it sounds the alarm on what appears to be very problematic behavior at some journals. We would like to make a number of suggestions for revision.

As detailed in our specific comments, we recommend that the authors make it more clear that many of the terms and concepts described in the manuscript have only appeared in their previous non-refereed work, including a preprint. We should stress that we encourage citing such sources, but we also think it is important to be transparent about their status and origin. 

We would also strongly recommend that the journal invite specialists in economics and statistics to review this paper, particularly the formulas used and subsequent findings. This material is outside of our expertise, but is critical to the paper's conclusions.

Introduction:

Paragraphs one and two of the Introduction appear to be a summary of, and borrow language from, the authors' preprint published in July 2020 (reference #1), with little else to support the subsequent hypothesis. It is also unclear from the preprint whether the authors limited their dataset to research articles, or included items such as editorials, which would skew figures for active editors.

Several phrases were taken directly from the preprint without suitable citation. (e.g. "We suggest that (1) a constantly high proportion of papers published by a group of authors, (2) particularly in the presence of relationships between the editors and these authors, and (3) publication of low-quality research, are key characteristics of a new type of illegitimate publishing entity, i.e. "self-promotion journals", which deserve further investigation.") The authors' passage that "In the field of academic publishing, the term 'self-promotion journal' was coined" is a good example of one that should be more transparent about the fact that the term was coined by the authors elsewhere and has not yet, best we can tell, been taken up by others.

The second paragraph of the introduction includes a second reference, to a blog post by a co-author from prior works, and should similarly have more transparent language attached. The authors state, referring to the post's findings: "Furthermore, many of these papers, with superficial or absent peer reviews, could be detected by the remarkably rapid turn-around, often within a week or less, between the dates recorded for submission and acceptance. This additional evidence of unethical editorial practice can only be obtained, however, in journals that report these dates for published manuscripts." We would recommend indicating how these reported dates were interpreted, because for some journals "acceptance date" means the date the paper was accepted *after* having gone through review. The manuscript may have been "accepted" but still required revisions which were done prior to publication. Similarly, a long time period between submission and acceptance does not indicate that a thorough, or even any, peer review was performed. And again, the nature of the publication (research, review, letter, editorial) was not clarified. Were comments related to a single study (e.g., "Authors Reply to Commenter, et al.") written by the same person considered separate articles?

The only other reference in the Introduction, found in the third paragraph, is to a statistical method (Gini) used primarily in economics for studying inequities in resource/income distributions. As noted in our general remarks, we would recommend that the journal seek an reviewer with this kind of expertise.

In the third paragraph, the authors state: "These convergent analyses from different fields suggest that the Percentage of Papers by the Most Prolific author (PPMP) is a simple measure that can be used as a red flag to identify journals that are suspected of biased editorial decision-making - what we now term 'nepotistic journals.'" This suggests that there is ample proof that a PPMP is a) "a simple measure" and b) "can be used as a red flag to identify journals that are suspected of biased editorial decision-making." We would recommend that the authors stress in their language that this is not yet a validated measure. (Consider too that journals starting up may have a disproportionately small author list, until well-known enough to attract a variety of new authors.)

Methods: 

Data extraction: 

1. "by the NLM for journals in the MEDLINE database" (emphasis ours): Did the authors differentiate between indexed in NLM, PubMed Central, and Medline? The confusion about the differences in terms NLM uses is of course not the fault of the authors, but some clarity would be helpful.

2. What did the authors do about journals having 2 or more Broad Subject Terms (BST). Is there unconsidered duplication within the single BST terms, or was the journal list just considered as a whole without separation per BST?

3. What are considered "published papers"? There are plenty of non-research papers indexed - including letters, editorials, and even tables of contents. What means (if any) of exclusion were applied for the bottom limit of 50 publications. This is especially important since the authors needed to revise their protocol to exclude "articles without an author name" - which would be quite an anomaly if concentrating on research and/or review papers.

4. How did the authors find all the articles per author? Was it through a NLM search via Pubmed or through the individual journal's separate table of contents? Not all journals list all articles within Pubmed - how was the comprehensiveness of the article check ensured?

4. "Publications reprinted in several journals (e.g., PRISMA statements published in 6 different journals to promote dissemination) did not receive special treatment, and no correction was applied, as each article was only examined in relation to its publication journal." How many of these cases occurred? Did the authors examine whether excluding these reprints affected any of the findings? It does not seem safe to assume that it would have no effect.

Index Calculation:

1. What did the authors do to confirm that similar/same names were the same person, or that different names were really different people (i.e., referring primarily last name/first name swaps - those using somewhat different names are likely rare, although they do exist). 

2. "When there was more than one author with the same largest number of published articles, they were all considered as the "most prolific" authors." If "most" is to be applied to more than one, perhaps a better term applies. 

3. Use of the Gini index should be assessed by a reviewer familiar with its use and application. 

4. "For the primary analysis, these two outcomes (PPMP and Gini) were computed for all papers, and for a sensitivity analysis they were computed only for papers labelled as 'journal articles' (using the NCBI publication type)." Please explain the difference between "all papers" and for "journal articles." 

Description of outlier journals

1.The term "outlier" could be misleading, as clearly this is referring to a specific group of journals created by specific measured behaviors rated along a particular scale, and "outliers" typically refer to statistical anomalies not generally expected to be part of a group.

2. We will defer to the statisticians in evaluating the true randomness of the sample. However, first alphabetizing journal names (with or without "The" in the title?), then selecting a seed set of 42 suggests "pseudo-randomization," not true randomization. Consider also that journals may avoid names/titles starting with letters towards the end of the alphabet and it becomes even less truly random.

Data analysis

1. See notes elsewhere about obtaining a review from someone with appropriate expertise in statistics or economics.

Protocol Changes

1. "because some articles unexpectedly lacked author names," again requires clarification as to the types of publications included as "article."

2. "The 3-week threshold used to describe publication lag as being suggestive of unduly rapid or absent peer review was not initially specified in our protocol, and was added for descriptive purposes." What is the basis of using 3 weeks as an arbitrary delineator? Is it just an assumption, or is there literature to show a relationship between 3 weeks and the quality of peer review? If so, include a reference. If not, say so and provide a justification for the time limit.

Results:

Journal Selection and Description

1. With such a broad range of numbers of articles published, might there not be a masking effect in journals with higher rates of publishing - causing lower-publishing-rate journals to show a greater number of PPMPs? Would it not have been more effective to break the journals into groups based on rates of publishing?

Description of the Indices

1. Statistics- defer to a specialist for review.

2. Slightly more than half the sample did not provide submission/acceptance rates. This suggests it is problematic to draw conclusions about "publication lag" calculations and comparisons

.

Description of outliers and identification of nepotistic journals

1. "206 based on the PPMP and the Gini index considered separately, and 68 based on both indices." If the PPMP and the Gini index each identified 206 different journals, and only agreed on 68, can the authors comment on the significant difference?

2. "The main characteristics of the 100 randomly selected outlier journals are presented" A statistician should confirm random vs pseudo-random. Re: the discussion of journals "reported in English." This is somewhat misleading since by nature the NLM is skewed towards English language journals.

3. "We identified errors in author identification for one journal, MMW-Fortschritte der Medizin (an outlier on the Gini index), where the most prolific 'author' was named 'Red', which seems to be a diminutive for 'Redaktion', possibly encompassing several physical individuals." Not sure how this would be considered "an error"? "Redaktion" appears to be German for "Editorial Staff." So these would more appropriately belong to the "unnamed" articles?

"4. When 'Red' was ruled out as a valid author name, the next most prolific author was however a member of the editorial board,…" For a journal with a highly active editorial board as demonstrated by the number of "Red" articles, would this be unusual or unseemly? Might it be possible that MMW-Fortschritte der Medizin has its own publishing emphasis?

5. How did the authors confirm that the same-named authors were indeed the same person? Did they cross-check affiliations, or just assume the same name in the same field would be the same person?

Discussion:

"In this comprehensive survey of 5 468 biomedical journals, we characterized several features of editor-author relationships among which were the following: (i) article output was sometimes dominated by the prolific contribution of one author or a group of authors, (ii) time lags to publication were in some instances shorter for these prolific authors and (iii) prolific authors were typically members of the journal's editorial board." Criteria for "sometimes" and "typically" are unclear. Additionally, as pointed out previously, the publication lag data was for less than half the journals considered in the sample, but was compared against figures for all the journals.

"We concluded that defining the top 5% nepotistic journals required the threshold to be set at up to 10.6% of articles published by the most prolific author." "for the purposes of this study" should be added. This study is insufficient for a general standard.

"In absolute terms, we believe it is reasonable to question the judgement of an editor where more than 10% of the published papers are authored by the same person." This may be reasonable, but without at least a validated spot-check of some of these papers, this statement appears to go well beyond what can be concluded from the findings. We note that a preprint of this manuscript has been the story of a news story in Science https://www.sciencemag.org/news/2021/02/journals-singled-out-favoritism that mentions the Didier Raoult oeuvre. Is there material there that could be cited?

"This suggests that a broader time-frame allowed for more occasional authors to be recruited while maintaining the regular authors, revealing the latent heterogeneity." Or revealed a beginning journal struggling to keep afloat long enough to attract a variety of authors and allow the "regular authors" to fade out to other publishing duties. Association is of course not causation.

"If a journal publishes a very large number of papers, it becomes increasingly implausible that a single prolific author could account for 10% or more of them, as there is a natural upper limit to how many papers any one individual can author." While this is not an unreasonable conclusion to draw, it should be noted that senior researchers who oversee multiple projects may end up as authors on many papers, which naturally ups their publication count.

"Our findings persisted when all articles were considered as well as when only 'journal articles' were considered (i.e. excluding articles explicitly referenced as editorial, correspondence or news articles), suggesting that editorials, correspondence and news, are not the only drivers of the indicators we explored." Did the authors confirm that the NLM labeling of type was accurate? PubMed relies on metadata from publishers, which has been shown to be incomplete or error-laden in a not insignificant number of cases.

"We should beware of assuming that a hyper-prolific author is necessarily engaged in questionable publishing practices: some people are naturally highly productive, and the speed with which good research can be completed is highly variable across research fields." Is a "hyper-prolific author" different to a PPMP? Here, those authors are given a pass, but a few paragraphs prior these authors cast aspersions on the editors for allowing such prolific behavior.

"It is also very important to complement the PPMP and the Gini index with the absolute numbers of papers authored by the most prolific authors, because some problematic journal behaviours could pass unnoticed when only these two indices are used." This suggests that the PPMP is not a "simple measure" as described elsewhere in the manuscript.

Limitations:

1. "but it may miss some finer points, especially concerning the quality of articles published in these journals." This is a significant limitation, as noted above, and deserves mention in the same breadth as other discussions of quality in the manuscript.

2. "Some journals are registered without broad terms, requiring a manual pick-up by the NLM." This is unclear. If the authors are referring to Medline and article indexings - the Publisher applies to Medline. No BSTs are required for the application.

3. "Similarly, because we restricted our search to journals publishing a minimum of 50 papers in the 2015-2019 period, we may have missed smaller journals with less professional editorial practices." Why is the assumption that smaller journals have less professional editorial practices? If this is referring to editorial staff, which is not the same as "editorial practices," that should be made clear.

4. "Importantly, our automated calculations carry a risk of inaccuracy as a result of homonymy…. Our analysis of the random sample of outliers enabled a disambiguation procedure consisting in inspecting qualitatively the most prolific authors. Only 1 out of 108 "most prolific" authors within a given journal was considered to be at being at high risk of homonymy." What was the disambiguation procedure? If only looking at the ones already considered to be prolific, how does this method ensure that others may miss such a determination due to slight differences in the names used? (For example, Erin Nicole Potts- Kant has published as Erin Potts, Erin Kant, Erin N Potts-Kant, etc. Others have swapped first, middle and last names, with minor spelling changes, depending upon the journal.) 

"Among these 108 authors, this procedure also enabled identification of the 6 most prolific authors, who were professional journalists for whom high productivity is of course not an indicator of any academic misconduct [emphasis ours], as they are professionals paid by the journal and not academics. The two proposed indicators, and their current calculation, should therefore not be used indiscriminately but could rather serve as a screening tool for potentially problematic journals that may then require careful exploration of their editorial practices." This also suggests the PPMP is not a "simple measure," and casts doubt on the earlier statement that "In absolute terms, we believe it is reasonable to question the judgement of an editor where more than 10% of the published papers are authored by the same person."

"The proposed indices could add transparency in the editorial decision-making and peer review process of any journal." We agree, provided the nuance and limitations are clearly reflected.

Reviewer #3:

The paper proposes and analyzes two interesting metrics for detecting possibly nepotistic practices in journals. Both, the PPMP and the Gini index, are easy to compute and to understand and they allow to identify outlier journals and authors than can be then double checked for editorial misconduct. The experimental design, data and code are available in OSF.

The analysis is rather descriptive and it would be great to see some statistical models in place (eg. when characterizing the profile of most prolific authors and of outlier journals). 

Although I understand the pragmatic decision of restricting the analysis of outlier journals to 100, I think the work should better cover all 480 journals.

The authors should more explicitly describe the actual contribution of some co-authors of this study.

COMMENTS FROM THE ACADEMIC EDITOR:

The report from the reviewers cover a lot of terrain in much detail. On top of this, there a couple of more general nature that I would like to give back as feedback that you might choose to use in a next version:

// take reader by the hand with a bit more explanation at the somewhat more complex or less widely known concepts. The figures are a good example - add perhaps 1-2 more sentences on what can be learned from the different figures, and how relate to each other. Another example is that some concepts are used, yet only explained in detail in the methods. That is relatively late due to the lay out format of PLOS Biol. This means that some aspects of the methods have to be introduced with a couple of extra words in the intro/results or discussion section. (e.g. gini index, "journal articles"

// journal age is only touched upon lightly in the narrative - I would like to point out fro personal experience with several new journals that when a new journal starts, there is active solicitation of papers from EiC to EB members to have enough copy in the starting up phase - What can be said about this phenomenon in relation to these indices?

// with regard to the suggestion of reviewer 3 to analyse a larger set of outlier journals I think it is important to note that while adding more observations will indeed increase precision, such a change will not affect the future decision of the PLOS Biol to publish or not publish.

// The openness on the used method (registered analysis, explicit mention of deviations in planned analysis) is great to see. If you decide to add to or change the analysis, feel free to note that these changes came through peer review in order to keep your level of "open reporting on methods used" intact. NB the level of detail on the selection procedure for outlier selection is recommendable. The seed number chosen is as obvious as right.

---

## [Decision Letter · Decision Letter 2]

5 Oct 2021

Dear Clara,

Thank you for submitting your revised Meta-Research Article entitled "‘Nepotistic journals’: a survey of biomedical journals." for publication in PLOS Biology. I've now obtained advice from one of the original reviewers and have discussed their comments with the Academic Editor. In addition, as advised in one of my emails, both this reviewer (reviewer #2) and the Academic Editor felt that the manuscript would benefit from further assessment of the specialised statistical methods used (new reviewer#4); please accept my apologies for the additional delay caused by this and some additional communication problems over the summer months.

Based on the reviews, we will probably accept this manuscript for publication, provided you satisfactorily address the remaining minor point raised by reviewer #4 and the following:

a) Please could you choose a more explicit and appealing title? We suggest the following ideas (assuming these are supported by your findings): "Editorial bias and nepotistic behaviour detected in 10% of biomedical journals" or "Nepotistic behaviour detected in 10% of biomedical journals" or "A survey of biomedical journals to detect editorial bias and nepotistic behavior." I should also say that we like to avoid punctuation in titles.

b) Please could you supply a blurb, as instructed in the submission form?

c) Please attend to the request from reviewer #4. Keep the Materials and Methods section in its current place, but address the reviewer's concern by adding a description of the Gini index earlier in the manuscript in order to help the reader appreciate the implications of its use more fully.

d) We note that your financial statement currently says “The author(s) received no specific funding for this work" - please can you confirm that this is correct?

e) We note that your supplementary Figures currently have rather confusing names. Please re-number them Figs S1, S2, S3, etc., provide legends for them, and cite them in the text.

f) We believe that the OSF deposition contains data and scripts sufficient to re-create all of the main and supplementary Figures. Can you confirm that this is the case? Please cite the location of the data clearly in each relevant main and supplementary Figure legend, e.g. "The data underlying this Figure may be found in https://osf.io/6e3uf/"

g) Please re-write the Abstract in the more verbose format that is standard for our journal. I also notice that something has gone wrong with the second sentence of the Abstract. 

We expect to receive your revised manuscript within two weeks. 

*Published Peer Review History*

*Early Version*

Best wishes,

Roli

Senior Editor,

rroberts@plos.org,

PLOS Biology

DATA NOT SHOWN?

REVIEWERS' COMMENTS:

Reviewer #2:

[identify themselves as Ivan Oransky and Alison Abritis]

We appreciate how seriously the authors have taken the suggestions of all of the reviewers, which indicates a high level of integrity in the work. The paper is much improved, and we are happy to recommend acceptance.

We would still recommend that the editors have a statistical expert review the methods, but we leave that to their discretion.

Reviewer #4:

Let me start by stating that in scientometrics/bibliometrics, the Gini-index is used more frequently, as a measure of concentration. I have personally used this as a measure of concentration with respect to field orientation, in other words, to what extent is a unit more mono-or interdisciplinary, and confront and correlate that with MRC research grant peer review assessments. 

With respect to this paper, a number of things strike me, in the first place the order of the sections in the manuscript. I would expect the Methods section before the Results section, which would mean that one would read about the used measures, among which the Gini-index, before reading results. This is unfortunate, as one now has to except or assume the meaning of the Gini-index, and how it work, and what it does. In the Methods section I think the presentation of the Gini-index is not sufficient, reading the section I did not get a clear view on how this index works, how it is calculated, how it is applied in the dataset that is collected, and also not how this affects the outcomes. I think the authors should add such a paragraph to the manuscript, as many readers of the paper will probably be not very familiar with the index, so a further explanation helps better understand the outcomes of the study. One would then also create a better understanding of how the Gini-index relates to the PPMP-indicator presented in the study, as well as the way these two measures are correlated, and how to interpret their correlation.

I liked the study and the paper, as it deals with an important research integrity issue, namely that of authorship, and everything that can go wrong there, which often remains invisible among all questionable research practices.

---

## [Editor Report · Decision Letter 3]

20 Oct 2021

Dear Clara,

On behalf of my colleagues and the Academic Editor, Bob Siegerink, I'm pleased to say that we can in principle offer to publish your Meta-Research Article "A survey of biomedical journals to detect editorial bias and nepotistic behavior" in PLOS Biology, provided you address any remaining formatting and reporting issues. These will be detailed in an email that will follow this letter and that you will usually receive within 2-3 business days, during which time no action is required from you. Please note that we will not be able to formally accept your manuscript and schedule it for publication until you have made the required changes.

PRESS: We frequently collaborate with press offices. If your institution or institutions have a press office, please notify them about your upcoming paper at this point, to enable them to help maximise its impact. If the press office is planning to promote your findings, we would be grateful if they could coordinate with biologypress@plos.org. If you have not yet opted out of the early version process, we ask that you notify us immediately of any press plans so that we may do so on your behalf.

Best wishes, 

Roli

Roland G Roberts, PhD 

Senior Editor 

PLOS Biology

rroberts@plos.org